# How to Reduce Physical Inactivity in School Context? A Systematic Review of the Concept of Body Practices

**DOI:** 10.3390/ijerph21091204

**Published:** 2024-09-11

**Authors:** Rodrigo Soto-Lagos, Tatiana Castillo-Parada, Luiz Pozo-Gómez, Pablo Romero-Álvarez, Gabriel Urzúa

**Affiliations:** 1Department of Management and Public Policies, Faculty of Administration and Economics, University of Santiago of Chile, Santiago 9170022, Chile; tatiana.castillo@usach.cl (T.C.-P.); luiz.pozo@gmail.com (L.P.-G.); pablo.romero.a@usach.cl (P.R.-Á.); 2Faculty of Psychology, UNIACC University, Santiago 7501277, Chile; gabriel.urzua@uniacc.cl

**Keywords:** physical inactivity, physical education, school, body practices, public policy

## Abstract

(1) Background: Physical inactivity is a recognized global problem, with schools identified by UNESCO and WHO as critical settings for intervention. Despite existing efforts, the prevalence of physical inactivity remains high. This study aims to explore the potential of the concept of body practices as a novel framework to address physical inactivity within school contexts, potentially offering an alternative to traditional intervention models. (2) Methods: This study carried out a systematic review of the concept of body practices to understand its use in the scientific literature. (3) Results: The results indicate that (a) the definition of the concept that the authors used is broad; (b) the problems they face are varied, such as the life cycle, negative emotions, gender, physical inactivity, well-being, and social problems; (c) the research methodologies are predominantly quantitative; (d) and the intervention in schools focuses on students, but not on the entire school community. (4) Conclusions: this concept has great potential for use in initiatives and/or policies that seek to address physical inactivity in the school context.

## 1. Introduction

Physical inactivity has been classified by the World Health Organization (WHO) as a global pandemic, with a particularly high prevalence among girls and women [1,2,3,4]. The COVID-19 pandemic has further exacerbated this issue, leading to a significant increase in physical inactivity worldwide [5]. Recognizing the severity of this problem, both the WHO and the United Nations Educational, Scientific and Cultural Organization (UNESCO) have emphasized the strategic importance of schools in addressing physical inactivity [6,7].

The predominant approach to combating physical inactivity in schools has largely been informed by the biomedical model, which has shaped public policy and intervention strategies [8,9,10,11]. However, scholars such as Lamelas [12] have criticized this model for its narrow focus on individual behavior, often neglecting the broader socioeconomic and cultural contexts that significantly influence physical activity levels [13].

One widely implemented school-based intervention model is the sport-for-health approach [14]. While this model has been effective in some contexts, it has also been criticized for its emphasis on competition, which can separate children, particularly girls, from participating in physical activities [15,16]. In response to these limitations, there has been a shift in public policy from a competitive sports model to a more inclusive sports-recreation model that integrates physical activity into daily life [17]. This transition highlights the importance of intersectoral collaboration in creating a comprehensive sports ecosystem that includes contributions from sectors such as health, education, and sports [10].

Given the limitations of the biomedical and sport-for-health models in effectively addressing physical inactivity, there is a growing need to explore alternative approaches that incorporate a broader range of factors influencing physical activity. One such approach is the concept of body practices, which expands the understanding of physical activity by integrating multidisciplinary perspectives [18,19,20]. Body practices are defined as manifestations of body culture, associated with the meanings that individuals assign to movement, gestures, and modes of body expression within specific cultural contexts [21].

The concept of body practices offers several advantages. It encourages a deeper understanding of how human movement is shaped by local realities and cultural discourses [19]. Moreover, it challenges the traditional meanings assigned to movement and body expressions, helping to explain why certain practices are embraced by some communities while resisted by others. This broader perspective includes not only sports and exercise but also other forms of physical activity, such as yoga, martial arts, gymnastics, and walking, which can be incorporated into school settings to offer more inclusive and culturally relevant physical education options.

In educational contexts, body practices have the potential to serve as alternatives to traditional physical education curricula, which may not always align with the interests or preferences of students [22]. Gamboa et al. [23] suggest that body practices represent an educational approach grounded in embodied experience, emphasizing the interaction between personal and collective histories within socio-cultural contexts.

Incorporating the concept of body practices into public policies and physical education programs could provide a more holistic approach to reducing physical inactivity. Despite efforts by states to address this issue, the WHO reports that inactivity levels remain alarmingly high, indicating the need for new strategies. Integrating body practices into the design and implementation of public policies could enhance their effectiveness and relevance, particularly in school settings.

This study aims to explore the potential of body practices as a conceptual framework for addressing physical inactivity. The following research questions guide this investigation:

How are body practices defined in scientific research?

What issues are addressed in research on body practices?

What methodological approaches are used in the study of body practices?

What are the key findings of research on body practices?

What future research directions are proposed in the literature?

## 2. Materials and Methods

This review adhered to the guidelines outlined in the Preferred Reporting Items for Systematic Reviews and Meta-Analyses (PRISMA) [24] and the Cochrane Handbook for Systematic Reviews of Interventions [25].

### 2.1. Design of the Literature Search Procedure

A comprehensive search of the scientific literature was conducted using three electronic databases: Web of Science, Scopus, and PubMed. The selection of these databases was strategically informed by their relevance and scope concerning our research objectives. Web of Science and Scopus are among the most extensive and multidisciplinary databases, recognized for their broad indexing of high-impact journals across various fields, including education and health sciences. PubMed, as a leading resource in health and biomedical sciences, was also included to ensure a thorough capture of the relevant literature. Given the interdisciplinary nature of our study, which intersects education, health, and the concept of body practices, these databases were selected to ensure a comprehensive and high-quality collection of research articles. Moreover, the rigorous indexing standards of these databases further guarantee the reliability and validity of the sources included in our review.

The search terms for this systematic review were meticulously chosen to reflect the central concepts of our study. Specifically, we employed the keywords “bodily practice” and “body practice” to capture literature that discusses the conceptual framework under investigation. These terms were paired with “school and education” to target studies within educational settings, and “health” to encompass research linking these practices to health outcomes. The search strategy was carefully crafted to ensure a comprehensive retrieval of studies aligned with our research objectives, utilizing Boolean operators to refine the search results. This approach enabled the identification of the most pertinent articles addressing the intersection of body practices, education, and health within school contexts (Figure 1).

The inclusion criteria were as follows: studies published within the last seven years; studies with at least the title available in English; and articles that address the concept of “body practice” and “bodily practice” in the context of education and health in schools.

### 2.2. Study Eligibility

The search procedure consisted of 3 phases (Figure 1): First, the mentioned electronic search was carried out, a stage in which all duplicate articles were eliminated. Second, an analysis of the title and abstract of each of the articles was carried out, filtering through the defined inclusion criteria, reaching 29 articles. Third, an identification of the main bibliographic elements of the articles was carried out; title, abstract, authors, country, sample, results, journal, and type of study (Table 1). Along with this, each of these articles was read in detail to make a final selection, after which, 10 articles were obtained. In the final stage of selection, one article was excluded because, although it met the inclusion criteria, it was an essay and documentary review of body practices in the 19th century, and its historical context was too distant from the rest of the articles. Thus, the final number of academic articles reached was 9.

## 3. Results

The analysis of the selected articles was carried out based on the five research questions mentioned above. As a result of the analysis, four categories were built. 

### 3.1. Definition of the Concept Body Practices

Following a comprehensive analysis of the articles addressing body practices in the school context, it is evident that none of the studies provide a precise definition of the concept, and the sources cited by the authors vary widely. Nevertheless, the articles generally refer to body practices as a concept that broadens the traditional understanding of physical activity and sports, encompassing a diverse array of bodily expressions and movements (as detailed in Table 2).

Upon closer examination, the articles identify various types of body practices, including structured sports such as soccer and volleyball, as well as less structured activities like dance. Additionally, the presence of activities associated with the arts—such as theater, circus arts, choir, music, and art therapy—is notable [26,27,28,29,30].

A significant finding from the analysis is the identification of a specialized area of study that focuses on “mind–body practices”. This concept refers to a range of disciplines and bodily expressions rooted in Eastern traditions, such as meditation, yoga, and mindfulness-based interventions. The articles that discuss body practices in schools often explore the interventions and effects of these practices [31,32,33,34]. Although these studies do not provide a comprehensive definition of body practices, they do offer specific definitions within this context: mindfulness is described as a practice that involves attending to the present moment with non-judgment, while yoga is defined as mindfulness in motion, characterized by deliberate, careful postures and a focus on breathing [35].

Regarding the school community participants involved in body practices, the articles predominantly focus on students. Only Carrol et al. [31] investigate a mindfulness-based intervention aimed at teachers. In terms of the timing of these practices within the school routine, all the reviewed articles mention after-school hours or extracurricular workshops as the primary settings. Physical education classes are referenced in two of the articles [27,30], while recess is examined in only one study [30]. 

### 3.2. Issues Addressed in Scientific Research

The review identified four primary dimensions in the research addressing body practices in school contexts. First, there is a significant focus on physical inactivity, which adversely affects children and adolescents at various stages of development [29,33]. This focus has prompted educators to prioritize self-regulation strategies, starting from elementary education [26]. Adolescence, in particular, is recognized as a critical phase due to the transition from elementary to secondary education, which brings changes in routines, academic demands, and physical, social, and emotional aspects [32].

Second, the emotional impact of body practices has been a subject of investigation, especially regarding students’ mental well-being and emotional health. Ezeh et al. [28] explored post-traumatic stress resulting from kidnapping experiences, highlighting the negative effects on students’ mental health.

Third, gender dynamics have been examined, particularly how body practices influence participation and reinforce or challenge gender roles in the school environment. Gender disparities significantly impact students’ access to and experience of school activities, including physical education. Bortoleto et al. [27] suggest that preferences for certain activities, such as circus arts, may be shaped by gender dynamics and historical pressures that condition disciplinary and performance practices. This highlights the importance of critically evaluating traditional physical education approaches, which often lead to unequal gender participation.

Fourth, social factors have been explored in relation to the implementation of body practices in schools. Andrade de Medeiros Moreira et al. [26] studied how childhood obesity and overweight are linked to social determinants such as urbanization, which affects the availability of recreational spaces, lack of parental and community involvement, and the sale of ultra-processed foods within schools. Additionally, Clarke et al. [32] noted that low socioeconomic status is correlated with poorer mental health outcomes, particularly among marginalized migrant communities, who face greater risks of victimization, housing instability, and discrimination.

Moreover, it is important to note that emotional challenges related to school environments affect not only students but also teachers. Carroll et al. [31] described teaching as a caregiving profession that entails significant mental and emotional exhaustion due to the heavy workload and the responsibility of addressing diverse and complex issues.

### 3.3. Methodological Approaches

The reviewed studies employed a range of methodological approaches. Four studies utilized quantitative methods [26,28,29,34], while three employed qualitative methods [27,30]. The studies by Carroll et al. [31] and Clarke et al. [32] adopted a mixed-methods approach.

Sample sizes varied across the studies, with the largest samples found in Andrade de Medeiros Moreira et al. [26], involving over a thousand students, and Soto-Lagos et al. [30], which included three schools with a comparable sample size. The other studies examined smaller samples, ranging from three students and three adults [33] to 470 students [28] and 187 students [32].

Various qualitative data collection methods were employed, including quasi-ethnography [30], focus groups [32], and semi-structured interviews [27,33]. Quantitative methodologies included the use of instruments such as the Five Facets of Mindfulness Questionnaire (FFMQ), Perceived Stress Scale (PSS), and Positive Mindset Index (PMI) in Carroll et al. [31]; the Self-Concept Multidimensional Scale, Perceived General Self-Efficacy Scale, Academic Self-Efficacy Scale, and International Physical Activity Questionnaire in Gasparotto et al. [29]; and the International Trauma Questionnaire in Ezeh et al. [28].

### 3.4. Implementation of Body Practices in Schools

Institutional support for implementing body practices in schools varies significantly depending on national policies and the specific characteristics of individual educational institutions. Bortoleto et al. [27] observed that private schools generally possess better resources, including teaching staff and infrastructure, which facilitate the integration of body practices into both extracurricular programs and the general curriculum.

The role of teachers in addressing physical inactivity is also highly valued by students. Soto-Lagos et al. [30] found that student motivation to engage in physical activities is strongly influenced by the enthusiasm and support of physical education teachers. Additionally, a well-structured lesson plan was shown to have a positive impact on student participation.

Research has demonstrated that incorporating body practices, such as yoga, into school programs positively affects students’ mental and physical health, particularly those who have experienced trauma. Similarly, dance has been effectively used to support Nigerian students who have undergone traumatic events, such as kidnappings, and have been diagnosed with post-traumatic stress disorder, leading to a reduction in distress symptoms [28].

Yoga, when compared to general physical exercise, has been found to be more effective in enhancing academic performance and reducing anxiety among adolescents, especially those with lower academic achievement [34]. Martin et al. [33] reported that yoga and other meditative activities also contribute to socio-emotional learning in children by improving attention and discipline through engaging and purposeful practice.

Mindfulness practices have similarly demonstrated benefits for children and adolescents, particularly those from low-income backgrounds who experience mental health challenges. Clarke et al. [32] found that adolescents reported reduced subjective distress after participating in a mindfulness intervention, which provided them with tools to manage academic stress and remain focused on the present.

For teachers, mindfulness-based programs have been shown to enhance health and well-being, leading to higher job satisfaction, reduced stress, and a lower likelihood of leaving the profession [31]. During an eight-week mindfulness intervention, teachers reported fewer negative reactions and an increased ability to remain calm and appreciate positive aspects of their lives.

## 4. Discussion

The analysis conducted in this study provides valuable insights into the application of body practices within school contexts. The findings align with Tremblay et al. [36], who argue that addressing physical inactivity requires considering social and cultural factors, which are integrated within the concept of body practices. Moreover, the results suggest that body practices can serve as an alternative to the sport-for-health model [14] for addressing physical inactivity globally.

These findings are consistent with Carvalho [27], who emphasizes the importance of incorporating materiality, social norms, social inequality, and context into the experience of movement and bodily expression. The reviewed studies also underscore the positive emotions generated by body practices and mind–body practices, aligning with the work of Olivera and Olivera [37] and Murillo [22].

While further research is needed [29] to explore the concept of body practices more comprehensively, the current findings offer a foundation for ongoing discussion. Positive evidence has emerged from interventions based on mind–body practices, such as yoga and mindfulness, particularly for students who have experienced trauma. However, there are concerns regarding the individualistic focus of these practices, which often overlook the social conditions that contribute to distress. Although these interventions promote well-being among students and teachers [31], it is essential to recognize that body practices not only enhance physical and subjective well-being but also promote social justice through a critical examination of the school environment. This highlights the need for expanded research that examines both private and public schools across diverse socioeconomic contexts.

Mindfulness, as a mind–body practice, has been recognized as an innovative approach to addressing subjective distress through strategies such as breathing and meditation. This practice has been extensively discussed in studies involving traumatized, low-income students and teachers [31,32]. However, the focus on the practice’s effects on individuals can sometimes obscure the broader cultural and contextual significance of movement. Consequently, while mind–body practices are part of the scientific literature on body practices, they do not always align fully with the broader conceptual framework.

Additionally, the limited duration of research on body practice interventions presents a challenge in thoroughly analyzing the effects of yoga and other practices on cognitive performance and anxiety reduction [34]. Future research should address these limitations by conducting longer-term studies with longitudinal observations and more representative student samples [29]. Despite these challenges, the growing interest in and support for body practices in educational contexts is promising.

Qualitative research methods are recommended to support the implementation of body practice programs in schools and to evaluate their impact on students’ subjective well-being [32]. These methods will allow for a deeper understanding of the benefits of body practices, capturing the experiences, perceptions, and meanings attributed by students, teachers, and other educational stakeholders.

It is important to highlight that the studies by Andrade de Medeiros Moreira et al. [26] and Gasparotto et al. [29] met the inclusion criteria by linking body practices with variables such as obesity, academic performance, and the amount/intention of physical exercise. Although these studies reference body practices, they predominantly adopt a biomedical perspective that overlooks a critical examination of the meanings these practices hold for individuals.

In this context, it is essential to underscore the concept of body practices, which emerges from a distinct epistemological and ontological standpoint related to bodily movement. On the one hand, this concept provides a more nuanced understanding of how human movement is shaped by local realities and cultural discourses (Fullagar, 2019). On the other hand, it expands beyond the realms of sports and exercise to encompass various forms of physical activity, drawn from both Eastern and Western traditions, such as yoga, martial arts, gymnastics, and walking. These activities can be incorporated into educational settings to foster more inclusive and culturally responsive physical education programs.

Thus, the use of the concept of body practices entails a stance that diverges from the biomedical approach. While it does not disregard the biomedical perspective, it extends beyond merely observing physiological processes in the study of bodily movement. Specifically, this concept integrates biological criteria with the social, cultural, historical, and political dimensions inherent in bodily movement.

The role of teachers is emphasized in the reviewed studies [27,30,31]. Teachers’ roles can be influenced by institutional and public policy levels, as the implementation of body practices in schools often depends on teachers’ initiative. Students are frequently perceived as passive participants, with interventions directed by teachers. However, Carroll et al. [31] offer a different perspective on the role of educators.

Institutional support is also crucial for the successful implementation of body practices [27,30,33]. This is in line with the ecological model proposed by Soto-Lagos et al. (2022), which argues that addressing physical inactivity requires considering multiple levels, including institutional factors, as individual-focused explanations are often insufficient.

These findings suggest future research directions, particularly the need to explore the meanings that body practices hold for children and adolescents, as part of a community and culture [27].

## 5. Conclusions

The analysis suggests that the concept of body practices introduces diverse perspectives and methodologies that extend beyond traditional frameworks, such as the biomedical model and individualistic interventions, which often isolate individuals from their social and cultural contexts.

Although the concept of body practices is not yet fully clarified in the literature, there is a discernible effort to broaden the understanding and application of this approach. This includes recognizing that body movement can be approached from perspectives that challenge the traditional view of sports as the primary means of promoting physical activity in educational settings.

## 6. Future Directions

This research underscores the need for a precise and rigorous definition of the concept of body practices in future studies, along with careful consideration of the ethical and political implications in the methodological design of such research.

## Figures and Tables

**Figure 1 ijerph-21-01204-f001:**
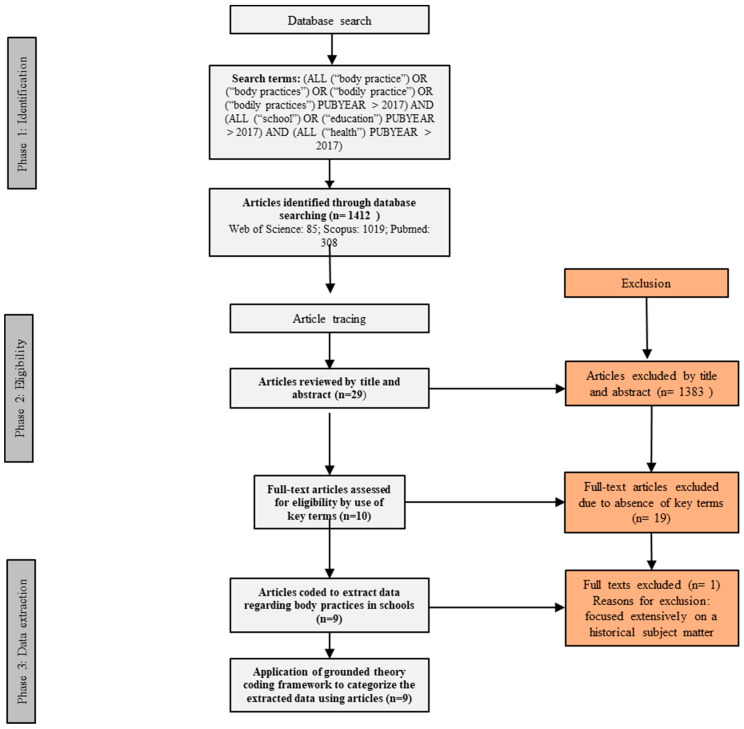
PRISMA flowchart for the systematic review of articles. The dark gray boxes indicate a process, the light gray boxes indicate an action related to a process, and the pink boxes indicate the articles excluded at each stage.

**Table 1 ijerph-21-01204-t001:** Bibliographic data of each article.

Authors	Year	Country	Sample	Method	Journal	Database
Soto-Lagos et al.	2023	Chile	3 educational institutions in the Valparaíso region in Chile	Quasi-ethnography	*Retos: nuevas tendencias en educación física, deporte y recreación*	Web of Science
Parajuli et al.	2022	India	89 girls between 12 and 14 years of age	Randomized clinical trial (RCT)	*Complementary Therapies in Clinical Practice*	Scopus
Clarke et al.	2022	EE.UU	187 Latino adolescents	Focus groups and mind–body Interventions	*Explore*	Scopus
Ezeh et al.	2022	Nigeria	470 students between 10 and 18 years old	Quasi-experimental design	*Journal of Pediatric Nursing*	Scopus
Carroll et al.	2022	Australia	18 teachers	Mixed-methods design and mind–body practice interventions	*Frontiers in Education*	Scopus
Martin et al.	2022	Australia	3 children and 3 adults	Semi-structured interviews	*Journal of Occupational Therapy, Schools, & Early* *Intervention*	Scopus
Bortoleto et al.	2020	Brasil	9 physical education teachers, 6 pedagogical coordinators, and 3 school principals	Semi-structured interviews	*Frontiers in Education*	Scopus
da Silva Gasparotto et al.	2020	Brasil	167 girls and 163 boys	Multidimensional Self-Concept Scale AF5	*Journal of Physical Education*	Scopus
Andrade de Medeiros Moreira et al.	2020	Brasil	1036 students from 25 public schools	Anthropometric measurement and questionnaires	*BMC Pediatrics*	Web of Science

**Table 2 ijerph-21-01204-t002:** Body practices present in the reviewed articles.

Body Practice	Number of Articles Where BP Is Present	Participants from the School Community Who Practice Them	Times When They Are Practiced
Artistic dance and dancing	4	Students	Recess, PE class, extracurricular workshop
Circus arts	1	Students	PE class and extracurricular workshop
Combat sport	1	Students	Extracurricular workshop
Performing arts—theater	2	Students	Extracurricular workshop
Soccer	1	Students	Recess, PE class, extracurricular workshop
Volleyball	1	Students	Recess, PE class, extracurricular workshop
Basketball	1	Students	Recess, PE class, extracurricular workshop
Rhythmic gymnastics	1	Students	Extracurricular workshop
Athletics	1	Students	Extracurricular workshop
Music choir	2	Students	Extracurricular workshop
Art therapy	1	Students	Extracurricular workshop
Yoga	4	Students and teachers	Extracurricular workshop
Mindful practice for teachers	1	Teachers	Extracurricular workshop

## Data Availability

The raw data supporting the conclusions of this article will be made available by the authors on request.

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
