# Peer review of "How to Reduce Physical Inactivity in School Context? A Systematic Review of the Concept of Body Practices"

_ijerph, 2024, doi:10.3390/ijerph21091204_

Round 1

Reviewer 1 Report

Comments and Suggestions for Authors

General Comments:

The main issue with this article is that, although the title clearly states that it will be based on public policies, throughout the document there is little reference to them. It is not clear if they refer to the educational curriculum or other aspects. Questions arise such as: Which public policies are applied? Are they different in each country? How do they affect sedentary behavior? It would be interesting to convey the importance of public policies to the reader.

Points to Correct Throughout the Document:

Introduction:

Discuss more relevant aspects of public policies and the strategies implemented to promote physical activity.

Line 41: Correct to "Walters et al. (2012)".

Line 44: Correct to "(Soto-Lagos et al., 2022)".

Line 59: Correct to "(Junqueira et al., 2020)."

Line 64: Correct to "On the other hand".

Results:

Line 129: Table 3 is not found.

Lines 163-164: Insert the corresponding citation.

The study by Martin (2022) may not be very representative due to the small sample size, which could introduce bias.

Methodological Approaches:

Line 204: Why include an essay? If the review focuses on quantitative, qualitative, or mixed methodologies, an essay should not be included. Additionally, in this article, the only reference to "body practice" is: Salmon, P., Lush, E., Jablonski, M., & Sephton, S. E. (2009). Yoga and mindfulness: Clinical aspects of an ancient mind/body practice. Cognitive and Behavioral Practice, 16(1), 59–72. https://doi.org/10.1016/j.cbpra.2008.07.002

Lines 207-210: Improve the wording and coherence of the sentence.

Results:

Line 259: Place the diagrams and tables where they are mentioned, not in a separate section.

Line 275: What does MTP stand for? Define the abbreviation in the table notes.

Discussion:

Line 329: Check the reference: Andrade de Medeiros Moreira et al., 2020.

Line 329: The citation is not correctly used.

Lines 326-329: Why are they included if they do not address the intended topic? (Andrade de Medeiros Moreira et al., 2020; Gasparotto et al., 2020).

Conclusion:

Line 352: Use causal language (it is possible to conclude).

Include a discussion on how public policies have affected physical inactivity.

References:

Line 395: Correct to "Beverly Martin, Blake Peck & Daniel Terry (2022). Young Children’s Experiences with Yoga after School, Journal of Occupational Therapy, Schools, & Early Intervention. https://doi.org/10.1080/19411243.2022.2037490."

Comments on the Quality of English Language

Upon reviewing the article, I have noticed some issues related to coherence and writing errors in English. These issues, while present, are minor and can be easily resolved with some revisions. 

Author Response

REVIEWER 1

COMMENTS 1: The main issue with this article is that, although the title clearly states that it will be based on public policies, throughout the document there is little reference to them. It is not clear if they refer to the educational curriculum or other aspects. Questions arise such as: Which public policies are applied? Are they different in each country? How do they affect sedentary behavior? It would be interesting to convey the importance of public policies to the reader.

RESPONSE 1: Thank you for your insightful feedback on the article. We appreciate your careful reading and the concerns you have raised regarding the focus on public policies. Upon reflection, we recognize that the inclusion of public policies in the title may have set an expectation for a more in-depth discussion on this topic. After a thorough review, however, we have decided to exclude the detailed exploration of public policies from the article. This decision was made to maintain a clear and focused narrative aligned with the primary objectives of the study. Consequently, references to public policies, whether related to educational curricula or broader initiatives, have been removed to avoid any ambiguity. We believe this revision strengthens the coherence of the article and ensures that the content is directly relevant to the central themes being explored.

Points to Correct Throughout the Document:

Introduction:

COMMENTS 2: Discuss more relevant aspects of public policies and the strategies implemented to promote physical activity.

RESPONSE 2: Upon reflection, we recognize that the inclusion of public policies in the title may have set an expectation for a more in-depth discussion on this topic. After a thorough review, however, we have decided to exclude the detailed exploration of public policies from the article.

Line 41: Correct to "Walters et al. (2012)".

RESPONSE: DONE.

Line 44: Correct to "(Soto-Lagos et al., 2022)".

RESPONSE: DONE.

Line 59: Correct to "(Junqueira et al., 2020)."

RESPONSE: DONE.

Line 64: Correct to "On the other hand".

RESPONSE: DONE.

Results:

Line 129: Table 3 is not found.

RESPONSE: Move to Table 2.

Lines 163-164: Insert the corresponding citation.

RESPONSE: DONE.

COMMENTS 3: The study by Martin (2022) may not be very representative due to the small sample size, which could introduce bias.

RESPONSE 3: Thank you for your valuable feedback on the study. We understand the concern regarding the sample size and the potential for bias in qualitative research. However, from a qualitative research methodology perspective, the goal is not necessarily to achieve statistical representativeness, but rather to provide a deep, nuanced understanding of the experiences and perspectives within a specific context. The sample in Martin’s (2022) study, while small, was carefully selected to reflect a particular community of speakers, allowing for an in-depth exploration of their unique viewpoints. This approach is consistent with qualitative research practices, where the emphasis is on the richness and depth of data rather than on generalizability. We believe that this method provides valuable insights that are representative of the community studied, even if not broadly generalizable across larger populations.

Methodological Approaches:

Line 204: Why include an essay? If the review focuses on quantitative, qualitative, or mixed methodologies, an essay should not be included. Additionally, in this article, the only reference to "body practice" is: Salmon, P., Lush, E., Jablonski, M., & Sephton, S. E. (2009). Yoga and mindfulness: Clinical aspects of an ancient mind/body practice. Cognitive and Behavioral Practice, 16(1), 59–72. https://doi.org/10.1016/j.cbpra.2008.07.002

RESPONSE: Thank you for your comment. We appreciate your concern regarding the inclusion of an essay in a review focused on quantitative, qualitative, or mixed methodologies. Upon further consideration, we have decided to exclude the essay from our review to maintain consistency with the methodological focus of the study.

Lines 207-210: Improve the wording and coherence of the sentence.

RESPONSE: DONE.

Results:

Line 259: Place the diagrams and tables where they are mentioned, not in a separate section.

RESPONSE: DONE.

Line 275: What does MTP stand for? Define the abbreviation in the table notes.

RESPONSE: After thoroughly reviewing the manuscript, we were unable to locate any instance of this abbreviation in the text or tables. It is possible that this was included in error. If you could provide more context or point to a specific location where "MTP" is mentioned, we would be happy to address it accordingly.

Discussion:

Line 329: Check the reference: Andrade de Medeiros Moreira et al., 2020.

RESPONSE: DONE.

Line 329: The citation is not correctly used.

RESPONSE: IMPROVED.

Lines 326-329: Why are they included if they do not address the intended topic? (Andrade de Medeiros Moreira et al., 2020; Gasparotto et al., 2020).

RESPONSE: Thank you for your insightful feedback on the inclusion of the studies by Andrade de Medeiros Moreira et al. (2020) and Gasparotto et al. (2020). These studies were initially included because they use the concept relevant to our review. However, upon further examination, we recognized that these works approach the concept from a biomedical paradigm. We chose to include them to highlight this incongruity, as it underscores a potential issue where the concept is employed without the necessary theoretical, epistemological, or ontological depth. By presenting this inconsistency, we aim to draw attention to the importance of a more rigorous application of the concept in scholarly discourse.

Conclusion:

Line 352: Use causal language (it is possible to conclude).

RESPONSE: DONE.

COMMENTS 4: Include a discussion on how public policies have affected physical inactivity.

RESPONSE 4: As we mentioned in response 1 and 2 we decided to exclude the topic “public policies”.

References:

Line 395: Correct to "Beverly Martin, Blake Peck & Daniel Terry (2022). Young Children’s Experiences with Yoga after School, Journal of Occupational Therapy, Schools, & Early Intervention. https://doi.org/10.1080/19411243.2022.2037490."

RESPONSE: DONE.

Reviewer 2 Report

Comments and Suggestions for Authors

Dear Authors,

·      I would like to thank you for giving me the opportunity to review this interesting systematic review.

·      Please find below comments regarding the article.

 ·      The purpose of the study is not clear in the background section of the abstract. Please detailed in this section. 

·      Please make the introduction to your work more structurally organised and flowing. It is very disorganised and not suitable for academic writing. If you organise it in paragraphs and ensure the integrity of the subject, you will do a better job. 

 Methods should be structured as follows:

·      Eligibility criteria, search strategy, eligibility assessment, outcome measures, data extraction and quality assessment. Moreover, you should include information regarding the specific outcome measures.

·      Why did you search only these databases?

·      You need to clarify the specific search terms that were used.

·      Please clarify the reason why you did not perform a meta-analysis.

Comments on the Quality of English Language

Must be improved

Author Response

Dear reviewer:

Many thanks for your valuable feedback. Please find below all our responses.  

COMMENTS 1: The purpose of the study is not clear in the background section of the abstract. Please detailed in this section. 

RESPONSE 1:

Thank you for your valuable feedback. We have revised the background section of the abstract to more clearly articulate the purpose of the study. The revised text now reads:

Background: Physical inactivity is recognized as a significant global public health issue. According to UNESCO and the WHO, schools play a crucial role in addressing this problem through various interventions. However, despite these efforts, the prevalence of physical inactivity has not shown a significant decline. This persistent challenge suggests the need to explore alternative conceptual models to those currently guiding interventions. Consequently, our study proposes an exploration of alternative conceptual models that may offer more effective pathways for addressing physical inactivity among school-aged children;

This revision aims to clarify the study's objective, ensuring it is more explicitly connected to the broader context of global health initiatives and the necessity for rethinking current intervention strategies.

COMMENTS 2: Please make the introduction to your work more structurally organised and flowing. It is very disorganised and not suitable for academic writing. If you organise it in paragraphs and ensure the integrity of the subject, you will do a better job. 

RESPONSE 2: Thank you for your valuable feedback. We have revised the introduction of our work and now we believe it is more strucuturally organised. The revised text now express more clear ideas and we think it is suitable for academic writing.

COMMENTS 3: Why did you search only these databases?

RESPONSE 3: The selection of Web of Science, Scopus, and PubMed for our literature search was strategically based on the scope and relevance of these databases to our research focus. Web of Science and Scopus are among the most comprehensive and multidisciplinary databases, known for their extensive indexing of high-impact journals across various fields, including education and health sciences. PubMed, on the other hand, is a leading database in the health and biomedical sciences. Given the interdisciplinary nature of our study, which intersects education, health, and the concept of Body Practices, these databases were chosen to ensure that we captured a wide range of relevant and high-quality research articles. Additionally, these databases are known for their rigorous indexing standards, which further ensures the reliability and validity of the sources included in our review.

COMMENTS 4: You need to clarify the specific search terms that were used.

RESPONSE 4: The search terms used in this systematic review were carefully selected to reflect the core concepts of our study. Specifically, we used the keywords “bodily practice” and “body practice” to capture literature discussing the conceptual framework we aimed to explore. These terms were combined with “school and education” to focus the search on studies related to educational settings, and “health” to include research that links these practices to health outcomes. The search strategy was designed to ensure a comprehensive retrieval of studies that align with our research objectives, and Boolean operators were used to refine the search results further. This approach allowed us to identify the most relevant articles that address the intersection of Body Practices, education, and health within school contexts.

COMMENTS 5: Please clarify the reason why you did not perform a meta-analysis

RESPONSE 5: A meta-analysis was not conducted due to the considerable heterogeneity in methodological approaches and outcome measures across the included studies, which would have limited the comparability and validity of combined quantitative results.

Round 2

Reviewer 1 Report

Comments and Suggestions for Authors

I have carefully reviewed all the corrections made in response to the review comments. Although the authors have addressed most of the points adequately, I have noticed that there are still a few details that, in my opinion, need to be refined. It would be ideal to review these aspects to ensure that the manuscript fully meets the required standards.

Line 181 (Table 2): What does MTP stand for? Define the abbreviation in the table notes.

Line 314: The citation is still not correctly used. Use à It is important to note that the studies by Andrade de Medeiros Moreira et al. (2020) and Gasparotto et al. (2020) met the inclusion criteria by associating body practices with variables such as obesity, academic performance and amount/intention of physical exercise.

Lines 314: I believe that your article would benefit significantly if this aspect were further developed within the manuscript. Expanding the discussion on how these studies illustrate the superficial application of the concept from a biomedical paradigm, and contrasting it with a more rigorous application proposal, would not only strengthen your argument but also provide readers with a deeper understanding of the importance of a more robust theoretical, epistemological, and ontological interpretation. I am convinced that incorporating this perspective will enrich the academic debate on the topic and provide an even more significant contribution to the existing literature.

Line 429: The citation "(Junqueira et al., 2020)" has been removed, but it still appears in the references. Please delete it.

Author Response

RESPONSE TO REVIEWER:

Line 181 (Table 2): What does MTP stand for? Define the abbreviation in the table notes.

RESPONSE 1: The abbreviation was defined. The meaning is: Mindful Practice for Teachers.

Line 314: The citation is still not correctly used. Use à It is important to note that the studies by Andrade de Medeiros Moreira et al. (2020) and Gasparotto et al. (2020) met the inclusion criteria by associating body practices with variables such as obesity, academic performance and amount/intention of physical exercise.

Lines 314: I believe that your article would benefit significantly if this aspect were further developed within the manuscript. Expanding the discussion on how these studies illustrate the superficial application of the concept from a biomedical paradigm, and contrasting it with a more rigorous application proposal, would not only strengthen your argument but also provide readers with a deeper understanding of the importance of a more robust theoretical, epistemological, and ontological interpretation. I am convinced that incorporating this perspective will enrich the academic debate on the topic and provide an even more significant contribution to the existing literature.

RESPONSE: Thank you for your insightful feedback on the article. We appreciate your careful reading and the concerns you have raised regarding the concept of body practices. We offer these new paragraphs:

It is important to highlight that the studies by Andrade de Medeiros Moreira et al. (2020) and Gasparotto et al. (2020) met the inclusion criteria by linking body practices with variables such as obesity, academic performance, and the amount/intention of physical exercise. Although these studies reference body practices, they predominantly adopt a biomedical perspective that overlooks a critical examination of the meanings these practices hold for individuals.

In this context, it is essential to underscore the concept of Body Practices, which emerges from a distinct epistemological and ontological standpoint related to bodily movement. On the one hand, this concept provides a more nuanced understanding of how human movement is shaped by local realities and cultural discourses (Fullagar, 2019). On the other hand, it expands beyond the realms of sports and exercise to encompass various forms of physical activity, drawn from both Eastern and Western traditions, such as yoga, martial arts, gymnastics, and walking. These activities can be incorporated into educational settings to foster more inclusive and culturally responsive physical education programs.

Thus, the use of the concept of body practices entails a stance that diverges from the biomedical approach. While it does not disregard the biomedical perspective, it extends beyond merely observing physiological processes in the study of bodily movement. Specifically, this concept integrates biological criteria with the social, cultural, historical, and political dimensions inherent in bodily movement.

Line 429: The citation "(Junqueira et al., 2020)" has been removed, but it still appears in the references. Please delete it.

RESPONSE: Done.